# Double p52Shc/p46Shc Rat Knockout Demonstrates Severe Gait Abnormalities Accompanied by Dilated Cardiomyopathy

**DOI:** 10.3390/ijms22105237

**Published:** 2021-05-15

**Authors:** Bradley Miller, Tatiana Y. Kostrominova, Aron M. Geurts, Andrey Sorokin

**Affiliations:** 1Department of Medicine, Medical College of Wisconsin, Milwaukee, WI 53266, USA; bmiller@mcw.edu; 2Department of Anatomy, Cell Biology and Physiology, Indiana University School of Medicine-Northwest, Gary, IN 46408, USA; tkostrom@iun.edu; 3Department of Physiology, Medical College of Wisconsin, Milwaukee, WI 53266, USA; ageurts@mcw.edu

**Keywords:** Shc signaling, sciatic nerve, dystrophin

## Abstract

The ubiquitously expressed adaptor protein Shc exists in three isoforms p46Shc, p52Shc, and p66Shc, which execute distinctly different actions in cells. The role of p46Shc is insufficiently studied, and the purpose of this study was to further investigate its functional significance. We developed unique rat mutants lacking p52Shc and p46Shc isoforms (p52Shc/46Shc-KO) and carried out histological analysis of skeletal and cardiac muscle of parental and genetically modified rats with impaired gait. p52Shc/46Shc-KO rats demonstrate severe functional abnormalities associated with impaired gait. Our analysis of p52Shc/46Shc-KO rat axons and myelin sheets in cross-sections of the sciatic nerve revealed the presence of significant anomalies. Based on the lack of skeletal muscle fiber atrophy and the presence of sciatic nerve abnormalities, we suggest that the impaired gait in p52Shc/46Shc-KO rats might be due to the sensory feedback from active muscle to the brain locomotor centers. The lack of dystrophin in some heart muscle fibers reflects damage due to dilated cardiomyopathy. Since rats with only p52Shc knockout do not display the phenotype of p52Shc/p46Shc-KO, abnormal locomotion is likely to be caused by p46Shc deletion. Our data suggest a previously unknown role of 46Shc actions and signaling in regulation of gait.

## 1. Introduction

Shc proteins function as adaptor proteins participating in the formation of multiunit signaling complexes and enabling a variety of signal transduction pathways [1]. Three mammalian Shc genes have been identified: *Shc1*, *Shc2*, and *Shc3*. *Shc1* is ubiquitously expressed, while expression of *Shc2* and *Shc3* is limited to the nervous system, where their products contribute to the development of subpopulations of both sympathetic and sensory neurons [2]. There are three members of Shc family of adaptor proteins encoded by *Shc1* gene, namely p66Shc, p52Shc, and p46Shc. They are known to execute distinctly different actions in cells, resulting in dissimilar functions in organisms [3,4,5]. Isoforms p46Shc and p52Shc arise from the use of alternative translation initiation sites within the same transcript, whereas p66Shc contains a unique N-terminal region and is believed to be generated by activation of alternative promoter. Due to its ability to be translocated to the mitochondria under stress conditions and facilitate production of ROS, p66Shc is generally considered to act as a sensor of oxidative stress [6]. p52Shc links receptor tyrosine kinases and G-protein coupled receptors with activation of Ras [7,8], while p46Shc was shown to inhibits thiolase and lipid oxidation in the mitochondria [5]. Whereas p66Shc knockouts were characterized in both mice and rats [9,10], much less is known about the consequences of p52Shc and p46Shc knockouts. We have recently demonstrated that even though genetic ablation of the p52Shc isoform significantly attenuated rat mammary tumor formation, p52Shc knockout rats (p52Shc-KO) do not display any phenotypic differences when compared with wild type rats [11]. It was reported that p46Shc isoform is up-regulated in skeletal muscle and spinal cord of 30-month-old rats [12]. Since specific p46Shc knockout has not yet been achieved in vivo, the exact biological significance of p46Shc signaling remains obscure.

In the current study, we characterized double p52Shc/p46Shc knockout rats (p52Shc/p46Shc-KO), which were easily recognized by severe gait abnormalities. Normal gait requires precise integration of the entire nervous system and gait abnormalities are often caused by the underlying neurologic condition [13]. Accordingly, we investigated p52Shc/p46Shc knockout rats for the neurological causes of abnormal gait. Gait disorders are common, they are sources of disability, and contribute significantly to morbidity through falls [14]. Uncovering the molecular mechanisms underlying the neurologic origins of gait disorders could offer an opportunity for effective therapeutic intervention [13].

## 2. Results

### 2.1. Generation of Rat Strains Deficient in Shc Isoforms

Five transgenic animals were established with CRISPR/Cas9-mediated targeted editing of the region of rat *Shc1* gene containing ATG start codon of p52Shc isoform on the genetic background of Dahl SS rats (Figure 1). These distinct, genetically modified rat strains either contained deletion of 5 bp, which resulted in predicted p66Shc-KO with p52Shc unaffected (m1 in Figure 1), deletion of 35 bp with predicted knockout of both p66Shc and p52Shc (m2 in Figure 1), deletion of 14 bp with similar predicted knockout of both p66Shc and p52Shc (m3 in Figure 1), or either 9 bp, or 6 bp deletions with resulting knockout of p52ShcKO along with 3 aa or 2 aa deletions in p66Shc, respectfully (m4 and m5 in Figure 1). Western blot analysis of genetically modified rat strain m4 revealed that both p52Shc and p46Shc isoforms are not expressed, whereas p66Shc expression is unchanged (Figure 1). The lack of p46Shc isoform is likely to be the consequence of alteration of initiation of alternative translation caused by the introduced 9 bp deletion.

### 2.2. p52Shc/p46Shc-KO Rats Demonstrate Severe Gait Abnormalities

Intercrosses of heterozygous parents resulted in a reduced number of homozygous p52Shc/p46Shc-KO animals, representing 10.2% (25 out of 246) of viable progeny (*p* < 0.05, chi-squared test). While neither p66Shc-KO, nor p52Shc-KO rats display any phenotype differences (body size, kidney weight, appearance, and breeding) from parental SS rats [9,11], surviving double p52Shc/46Shc-KO rats demonstrated severe gait abnormalities in 11 days old pups through three months of age (Appendix A). Gait analysis was quantified by analyzing the total paw print (Figure 2). The abnormal gait of p52Shc/p46Shc knockout rats resulted in increased paw print area compared to wild type or heterozygous p52Shc/p46Shc littermates due to the inability to maintain normal body position when walking. 

### 2.3. Total Body and Muscle Mass of p52Shc/p46Shc-KO Rats

The inability to maintain a normal gait could be the consequence of decreased muscle mass and muscle strength. In order to avoid any possible interference from the variations of breeder maintenance, we compared homozygous p52Shc/46Shc-KO and wild type littermates. Three rats per each group were analyzed. Total body weight of homozygous p52Shc/46Shc-KO rats (176.7 ± 4 g) was slightly lower than the wild type littermates (201 ± 9 g), but the difference did not reach statistical significance. To determine whether there was a difference in the muscle weight the following muscles were compared: Gastrocnemius (GTN), tibialis anterior (ATB), plantaris (PLN), extensor digitorum longus (EDL), and soleus (Sol) (Figure 3). Unadjusted and total body weight adjusted values are presented. Unadjusted weight of GTN muscle was statistically lower in homozygous p52Shc/p46Shc-KO rats when compared with the wild type littermates (Figure 3). There was no statistical difference when unadjusted weights of the other muscles were compared. The comparison of muscle weights after adjustment for the total body weight has not revealed any statistically significant differences (Figure 3). In heterozygous p52Shc/p46Shc-KO littermates, total body weight and muscle weight were not different from the wild type littermates (data not shown).

### 2.4. Histological Analysis of Skeletal Muscle of p52Shc/46Shc-KO Rats

Due to the decreased muscle weight of p52Shc/p46Shc-KO rats, GTN muscle was selected for histological evaluation. The comparison of muscle fiber outlines of white (glycolytic) and red (oxidative) areas of GTN muscle did not reveal profound differences between p52Shc/p46Shc-KO rats and wild type littermates (Figure 4A). There were no atrophic or damaged muscle fibers detected. Overall, muscle architecture looked similar as no fibrotic changes were observed (Figure 4A).

The analysis of muscle fiber type composition of red (oxidative) areas of GTN muscle and diaphragm muscle did not detect large differences between p52Shc/p46Shc-KO rats and wild type littermates (Figure 4B). There was no fiber type grouping present.

### 2.5. Analysis of p52Shc/46Shc-KO Sciatic Nerve Revealed Abnormalities

Immunostaining of the sciatic nerve cross-sections for neural cell adhesion molecule (NCAM) showed that homozygous p52Shc/p46Shc-KO rats have larger differences in nerve fiber diameters when compared with the wild type littermates (Figure 5A). In the wild type rats, nerve fibers have a uniform diameter and NCAM positive staining of axons is present in all fibers (Figure 5A). In sciatic nerves of p52Shc/p46Shc-KO rats, many large diameter nerve fibers lack NCAM axonal staining (Figure 5A). 

Similar differences were observed with immunostaining against plasma membrane calcium ATPase (PMCA) (Figure 5B). In p52Shc/46Shc-KO rats some areas of the sciatic nerves have large diameter nerve fibers that are missing PMCA axonal staining (Figure 5B). Many of the large diameter nerve fibers have very small diameter axons relative to the size of the myelin sheets (Figure 5B). In wild type rats, nerve fibers and axons have uniform diameter and axons are present in all fibers (Figure 5B). NCAM and PMCA were used to visualize axons since these are neuronal markers.

### 2.6. Analysis of p52Shc/p46Shc-KO Cardiac Muscle Revealed Features of Cardiomyopathy

Immunostaining for dystrophin showed that p52Shc/46Shc-KO rats have large areas with cardiomyocytes that are lacking specific staining (Figure 6). At the same time, these areas have intact wheat germ agglutinin (WGA) staining and cardiomyocytes have a normal size and shape. WGA is a lectin that stains connective tissue around cells. In skeletal muscle sections, we used WGA to visualize endomysium around muscle fibers. In sciatic nerve sections, we used WGA to visualize endoneurium surrounding myelin sheet and centrally located axon. In wild type rats, all cardiomyocytes displayed intact dystrophin staining and uniform cross-sectional area (Figure 6). It must be clarified that in this study, dystrophin was used exclusively as a marker to evaluate cardiomyopathy. Even though, it is well known that humans and rodents with dystrophinopathies have abnormal hearts, in our genetically modified rats, dystrophin immunostaining in skeletal muscle is normal, no dystrophinopathies, and no dystrophin mutations. It was reported in rats that early dystrophin loss in cardiomyocytes is coincident with the transition of compensated cardiac hypertrophy to heart failure [15]. Accordingly, dystrophin staining was used to test whether there are abnormalities in our genetically modified rats and dystrophin immunostaining disruption suggests dilated cardiomyopathy. We have only analyzed rats that were 2.5–3 months old, and at this age, we have not detected any cardiomyocyte atrophy or heart fibrosis. Future studies will be designed to clarify the relation between p46Shc signaling and the development of dilated cardiomyopathy.

## 3. Discussion

Genetically modified SS rats, which lack two Shc isoforms p52Shc and p46Shc (revealed by Western blot analysis), demonstrate severe gait abnormalities. Dilated cardiomyopathy, a clinically heterogenous disease, is characterized as left ventricular dilation and systolic impairment in the absence of common pathophysiologic conditions. The diagnosis and risk prediction of dilated cardiomyopathy are challenging because there is great heterogeneity in phenotype and genotype [16]. It is of note that the role of *Shc1* gene (also termed *ShcA*) in the regulation of cardiac structure and function was previously proposed. Moreover, myocardial-specific knock-in of murine ShcA phosphotyrosine binding domain resulted in the development of dilated cardiomyopathy [17] and motor behavior abnormalities [18]. In our experiments, p52Shc/p46Shc-KO rats also displayed some features of dilated cardiomyopathy. Loss of dystrophin staining can be considered as one of the markers of cardiac hypertrophy and early heart failure [15] and loss of dystrophin immunostaining was observed in enterovirus-induced dilated cardiomyopathy in humans [19]. In our experiments, wild type cardiomyocytes displayed prominent peripheral dystrophin expression whereas dystrophin is absent in large numbers of cardiomyocytes in the heart of p52Shc/p46ShcKO rats. Notably, disruption of dystrophin signaling complex was also reported in mice with conditioned deletion of ShcA PTB domain in heart [20]. At the same time, overall cardiac morphology was preserved in our mutant rats, no inflammation, fibrosis or apoptosis was observed in areas deficient in dystrophin immunostaining. This suggests that p52Shc/p46ShcKO rats at two months of age display early signs of heart failure associated with dilated cardiomyopathy.

The connection of p52Shc/p46Shc knockout with proximal muscle weakness is likely to be explained by the sciatic nerve abnormalities. Comparison of total body and muscle mass of p52Shc/p46Shc-KO and wild type littermates did not reveal any significant differences. There were no visible fiber size or fiber-type composition differences between gastrocnemius muscles of p52Shc/p46Shc-KO and wild type rats. Remarkably, p52Shc/p46Shc-KO rats had high size heterogeneity of the sciatic nerve fibers. Moreover, many myelin sheets did not have any axons. In contrast, WT nerve fibers were uniform in size and all myelin sheets had axons, however, alternative interpretations cannot be discounted. It has been reported that a lack of muscle control causing the ataxic gait in patients with Riley–Day syndrome is due to the loss of functional muscle spindles [21].

Body fat composition is linked to cardiovascular diseases and genome-wide association studies (GWAS) for the proportion of body fat distributed to the legs and trunk reports SNP in *Shc1* gene as one of body fat ratio-associated loci that have not previously been associated with an anthropometric trait [22]. It must be taken into consideration one criticism of GWAS is that most association signals reflect variants and genes with no direct biological relevance to the disease [23]. Furthermore, we observed severe gait abnormalities accompanied by dilated cardiomyopathy only in our homozygous p52Shc/p46Shc-KO rats, whereas heterozygotes were not different from the wild type rats. Notably, homozygote ShcA knockout mice die at embryonic day 11.5 [24]. 

p46Shc isoform is synthesized by employing an alternative initiation translation site using the p52Shc/p46Shc mRNA transcript [4]. One of the limitations of the current study is that the loss of p46Shc was not expected after CRISPR-mediated targeted editing of the p52Shc start codon. Nevertheless, it appears that the introduced deletion resulted in the efficient removal of p46Shc along with p52Shc. A possible effect of the introduced deletion upon mRNA stability of p52Shc/p46Shc transcript cannot be eliminated [25]. p46Shc function in mitochondria is to negatively regulate thiolase activity and reduction of p46Shc expression was reported to activate thiolase and increase lipid oxidation [5]. Although decreased p46Shc expression in mice was suggested to contribute to the lean phenotype and even healthy aging [5], our p52Shc/p46Shc-KO rats clearly display severe health problems, including dilated cardiomyopathy and abnormal gait. 

Sciatic nerve abnormalities in p52Shc/p46Shc knockout rats could be accompanied with pathological changes in the cerebrum or cerebellum. Detailed analysis of brain structures in our genetically modified rats is beyond the scope of this paper, which is the first report of consequences of double knockout of two Shc isoforms, not previously described in any system.

We have previously generated and characterized SS rats with targeted edits of *Shc1* gene, which either lacks isoform p66Shc [9] or isoform p52Shc [11]. The fact that none of these genetically modified rat strains displayed abnormal gait or dilated cardiomyopathy suggests that it is the ablation of particularly p46Shc-dependent signaling that plays the principal role in sciatic nerve abnormalities and the development of cardiomyopathy. The significance of the current study is in discovering the novel function for an important signaling molecule and establishing the basis for novel therapeutic approaches in the treatment of dilated cardiomyopathy.

## 4. Materials and Methods

### 4.1. Animals

All animal protocols were conducted per the National Institutes of Health Guide’s guidelines for the Care and Use of Laboratory Animals and reviewed and approved by the Institutional Animal Care and Use Committee at the Medical College of Wisconsin. Transgenic rats were developed on the background of the Dahl salt-sensitive strain SS/JrHsdMcwi obtained from the colony at the Medical College of Wisconsin. p52Shc knockout rats (SS-Shc1^em6Mcwi^; RGDID:124715482) were generated by pronuclear injection of a CRISPR-Cas9 plasmid (pX330) targeting the sequence CACTCAGCTTGTTCATGTCC into one cell SS (SS/JrHsdMcwi) rat embryos as previously described [26,27,28]. Founder animal’s DNAs were PCR amplified using primers forward 5′-GACCCATTCTGCCTCCTCTG-3′; reverse 5′-TATGCACTCACCCGAACCAA-3′ and genotyped by the Cel-1 assay [29], then candidate founders confirmed by Sanger sequencing. A single founder was identified harboring a 9-bp deletion (5′-GGACATGAA-3′, rn6 chr2: 188,748,893–188,748,901) overlapping the translational start site of the p52 isoform of Shc1. This founder was then backcrossed to the parental strain to establish a mutant colony. 

### 4.2. Harvesting Muscle Samples

Female rats (three per group) at two months of age were euthanized by the creation of a bilateral pneumothorax under isoflurane anesthesia. Skeletal muscles and heart were dissected, weighed, placed into a TBS medium (Triangle Biological Sciences, Durham, NC, USA), and frozen in cold isopentane. Muscle samples were stored at −75 °C for the subsequent analyses.

### 4.3. Gait Analysis

Rats (8- to 11-week-old) were pre-trained for acclimation to the tracking corridor. Front and hind paws were dipped in non-toxic ink and rats were allowed to walk through the corridor with blank white paper on the floor. The traces were digitally scanned and thresholded to the ink trace. The ink trace was reported as a percentage of the total trace area, defined by the distance to the final ink mark. 

### 4.4. Histochemical and Immunohistochemical Analysis

Samples were sliced with a cryostat, adhered to the Superfrost Plus microscopy slides and used for the immunostaining. After fixation with ice-cold methanol for 10 min sections were rinsed three times with Phosphate Buffered Saline (PBS). Then, sections were blocked for 30 min with PBS-0.05% Tween20 (PBST) containing 20% calf serum (PBST-S) at room temperature and incubated overnight at 4 °C with the solution of primary antibodies in PBST-S. The following primary antibodies were used: Mouse anti-slow MHC (clone A4.84) obtained from the Developmental Studies Hybridoma Bank (Iowa City, IA, USA), rabbit anti-NCAM (Chemicon International, Temecula, CA, USA), mouse anti-dystrophin (Novocastra Laboratories, Newcastle, UK), mouse anti-PMCA (ABCAM, Cambridge, MA, USA). Sections were rinsed three times with PBS and then incubated for one hour at room temperature with Cy3-conjugated anti-mouse or anti-rabbit antibody (Jackson ImmunoResearch Lab., West Grove, PA, USA). Co-staining of sections with fluorescein-conjugated wheat germ agglutinin (green, WGA-fluorescein; 1 mg/mL; Molecular Probes, Eugene, OR, USA) was used to visualize connective tissue around muscle and nerve fibers and cardiomyocytes, as previously described [30]. Nuclei were stained by incubation with a DAPI solution (Sigma, St. Louis, MO, USA) in PBST. The sections were examined and photographed with a Leica microscope. 

## Figures and Tables

**Figure 1 ijms-22-05237-f001:**
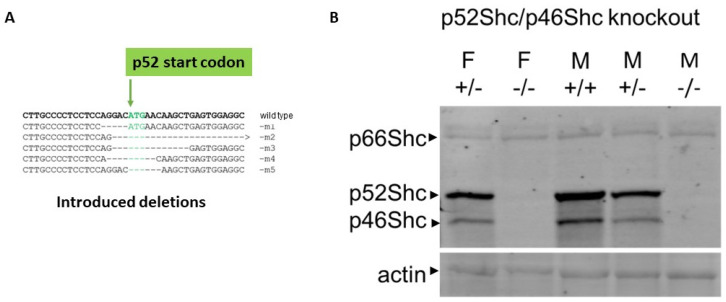
Generation and characterization of rats deficient in Shc isoform expression. (**A**) Introduced deletions into *Shc1* gene. The position of p52Shc start codon is indicated. (**B**) Protein expression of Shc isoforms in wild type (WT) and genetically modified rats. Expression of three Shc isoforms in renal tissues of wild type (+/+), heterozygote (+/−) and p52Shc/p46Shc knockout (−/−) rats is shown by Western blot with anti-Shc antibodies following SDS-PAGE. The position of Shc isoforms is indicated. Equal loading was verified by anti-β-actin antibodies.

**Figure 2 ijms-22-05237-f002:**
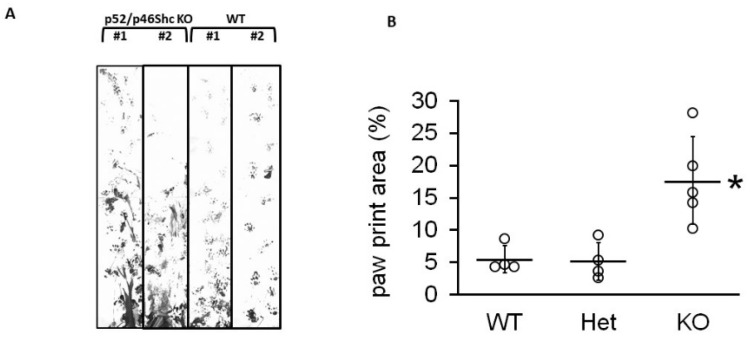
Gait analysis (**A**) Representative examples of gait profile were recorded for p52Shc/p46Shc-KO, two left lanes, and WT, two right lanes. In WT, a consistent step and stride pattern of paw prints was established by the end of the trace, whereas, in knockout animals, lower abdomen and/or hind paw drag created smeared traces between shorter, more irregular front paw prints. (**B**) Summary analysis of total paw print area. Het, rats heterozygous for both p52Shc/p46Shc-KO. Individual data points are shown, with horizontal lines designating mean ± standard deviation; * *p* = 0.007.

**Figure 3 ijms-22-05237-f003:**
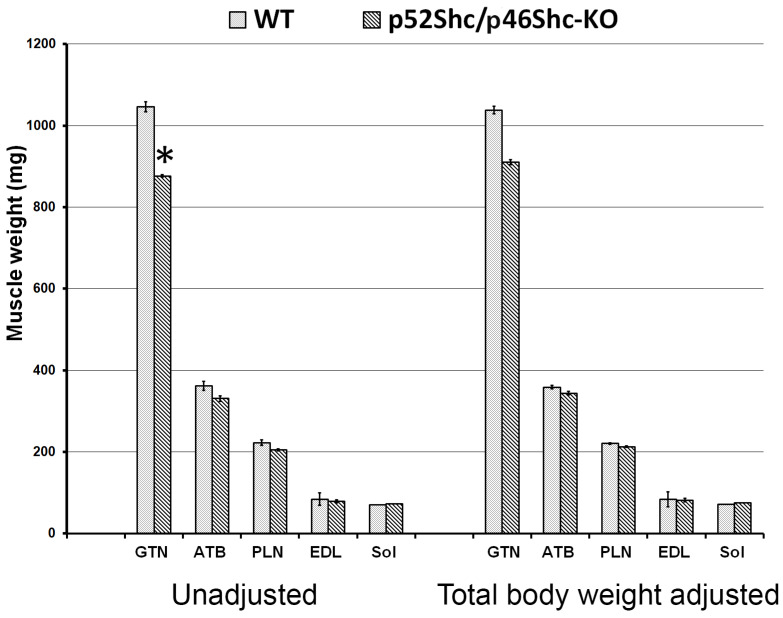
Skeletal muscle mass of p52Shc/p46Shc-KO rats and WT littermates. Absolute unadjusted and total body weight adjusted weight of GTN, PLN, EDL, and Sol muscles is presented. *: indicates a significant difference from the WT definition.

**Figure 4 ijms-22-05237-f004:**
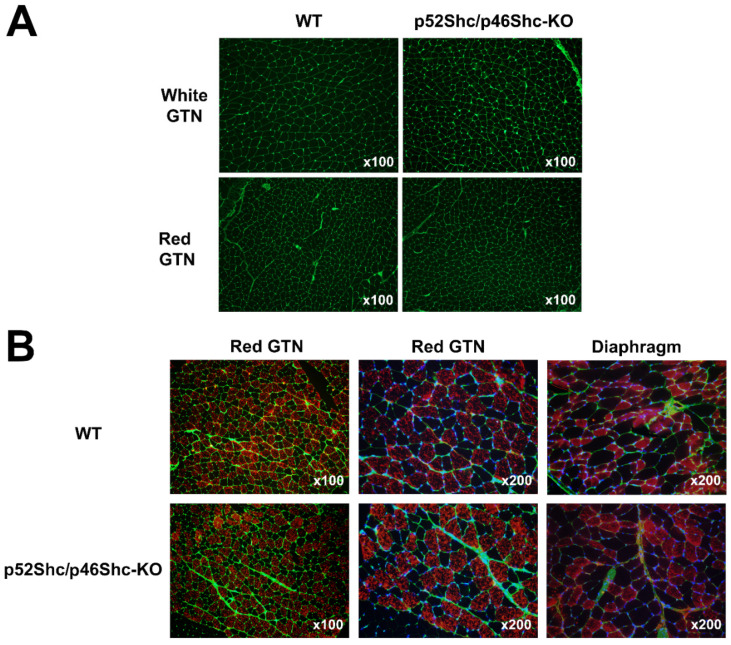
Comparison of GTN muscle and diaphragm morphology and fiber type composition of p52Shc/p46Shc-KO rats and WT littermates. (**A**) Cross-sections of white (glycolytic) and red (oxidative) areas of GTN muscle stained with WGA-fluorescein (green staining) are presented. (**B**) Cross-sections of red (oxidative) areas of GTN muscle and diaphragm stained with anti-slow MHC antibody (red staining) and WGA-fluorescein (green staining) are presented. DAPI (blue staining) was used to visualize nuclei.

**Figure 5 ijms-22-05237-f005:**
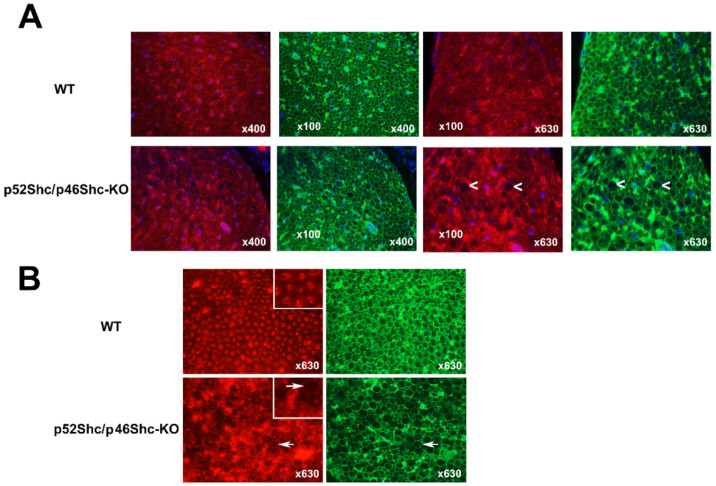
Analysis of sciatic nerve cross-sections of p52Shc/p46Shc-KO rats and WT littermates. (**A**) Cross-sections of sciatic nerve were stained with anti-NCAM antibody (red staining) and WGA-fluorescein (green staining). DAPI (blue staining) was used to visualize nuclei. White arrowheads indicate large myelin sheets that are missing axons in the cross-sections of p52Shc/p46Shc-KO rats. (**B**) Cross-sections of sciatic nerve were stained with anti-PMCA antibody (red staining) and WGA-fluorescein (green staining). White arrows indicate large myelin sheets that are missing axons in the cross sections of p52Shc/p46Shc-KO rats.

**Figure 6 ijms-22-05237-f006:**
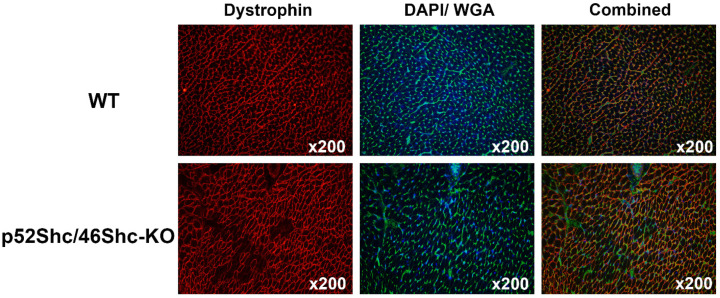
Analysis of cardiac muscle of p52Shc/p46Shc-KO rats and WT littermates. Cross-sections of heart were stained with anti-dystrophin antibody (red staining) and wheat germ agglutinin (WGA)-fluorescein (green staining). DAPI (blue staining) was used to visualize nuclei. Combined images have dystrophin, WGA-fluorescein and DAPI staining. Note large areas that are missing dystrophin staining in the heart section of p52Shc/p46Shc-KO rat.

## Data Availability

The data presented in this study are available in current article or Appendix A here.

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
