# Peer review of "Double p52Shc/p46Shc Rat Knockout Demonstrates Severe Gait Abnormalities Accompanied by Dilated Cardiomyopathy"

_ijms, 2021, doi:10.3390/ijms22105237_

Round 1

Reviewer 1 Report

The present manuscript, written by Bradley Miller and colleagues, is an original paper on the role of three isoforms: p46Shc, p52Shc and p66Shc of the adaptor protein Shc in rat physiology and susceptibility to various pathologies.

This is a well-planned, performed and written basic study. The study provides an interesting and worth exploring in future studies, novel potential mechanism involved in the development of dilated cardiomyopathy. Neither major nor minor concerns were identified.

Author Response

Reviewer 1 did not identify any neither major nor minor concerns.

Reviewer 2 Report

The research was well-designed and served purpose of the study. In this study, a double p52Shc/p46Shc knockout characterization of rats (p52Shc/p46Shc-KO) was done that was easily recognized by severe gait abnormalities.

The work is well-organized and the conclusions are adequately supported by the data. However, the following minor changings are required before publication in International Journal of Molecular Sciences.

  • Abstract need to more specific and organized.
  • Introduction need to more comprehensive with provision of some more data and relevant studies.

Author Response

Response to Reviewer 2 Comments

Point 1: Abstract need to more specific and organized.

Response 1: Abstract was re-written. All changes are indicated.

Point 2: Introduction need to more comprehensive with provision of some more data and relevant studies.

Response 2: Introduction was expanded, and new relevant studies were included. All changes are indicated.

Reviewer 3 Report

The investigated p52Shc/p46Shc knockout rats for the neurological causes of abnormal gait.

They showed significant anomalies of the sciatic nerve, the impaired gait, and the lack of dystrophin in some fibers in heart muscle, suggesting dilated cardiomyopathy.

This manuscript has remained several concerns.

Introduction

Normal “gate” requires precise integration of the entire nervous system and “gate” abnormalities are often caused by the underlying neurologic condition.

Were they typo?

Results

What was the number of rats used for figure 3?

2.5. Analysis of p52Shc/46Shc-KO sciatic nerve revealed abnormalities

What did NCAM and PMCA stand for?

The authors should explain it and describe in the manuscript.

The authors stated that immunostaining for dystrophin showed that homozygote p52Shc/46Shc-KO rats have large areas with cardiomyocytes that are lacking specific staining.

What was the relationship between dystrophin and Shc?

It is hard to understand how dystrophin were correlated with Shc.

What did WGA stand for?

The authors should explain it and describe in the manuscript.

The authors stated that p52Shc/46Shc-KO cardiac muscle revealed features of cardiomyopathy.

What points in p52Shc/46Shc-KO rats showed dilated cardiomyopathy?

If they had dilated cardiomyopathy?

Did they have heart failure, likely reflected to left ventricular ejection, left ventricular diastolic dimension, enlargement of heart, or cachexia?

It is hard to understand why p52Shc/46Shc-KO rats showed dilated cardiomyopathy.

The authors should clarify it and show more experimental data regarding cardiac involvement.

Discussion

The authors speculated that the connection of p52Shc/p46Shc knockout with proximal muscle weakness is likely to be explained by the sciatic nerve abnormalities.

Were there any pathological changes in the cerebrum or cerebellum?

Author Response

Response to Reviewer 3 points

Point 1: Normal “gate” requires precise integration of the entire nervous system and “gate” abnormalities are often caused by the underlying neurologic condition. Were they typo?

Response 1: Indeed, they were typo, which are now corrected. Lane 66.

Point 2: What was the number of rats used for figure 3?

Response 2: We used three rats per group for Figure 3. Clarification is included on lanes 122 and 299 (in Methods).

Point 3: What did NCAM and PMCA stand for? The authors should explain it and describe in the manuscript.

Response 3:  Explanation for NCAM (neural cell adhesion molecule) is added on lane 164, explanation for PMCA (plasma membrane calcium ATPase) is on lanes 171 and 172.

Point 4:  What was the relationship between dystrophin and Shc? It is hard to understand how dystrophin were correlated with Shc.

Response 4:  In this study dystrophin was used exclusively as a marker to evaluate cardiomyopathy. Even though, it is well known that humans and rodents with dystrophinopathies have abnormal hearts, in our genetically modified rats dystrophin immunostaining in muscle is normal, no dystrophinopathies, and no dystrophin mutations. It was reported, that in rats early dystrophin loss is coincident with the transition of compensated cardiac hypertrophy to heart failure. Accordingly, dystrophin staining was used to test whether there are abnormalities in our genetically modified rats and dystrophin immunostaining disruption suggests dilated cardiomyopathy.

Point 5: What did WGA stand for? The authors should explain it and describe in the manuscript.

Response 5: Explanation for WGA (wheat germ agglutinin) is added on lane 209.

Point 6: The authors stated that p52Shc/46Shc-KO cardiac muscle revealed features of cardiomyopathy. What points in p52Shc/46Shc-KO rats showed dilated cardiomyopathy? If they had dilated cardiomyopathy? Did they have heart failure, likely reflected to left ventricular ejection, left ventricular diastolic dimension, enlargement of heart, or cachexia? It is hard to understand why p52Shc/46Shc-KO rats showed dilated cardiomyopathy. The authors should clarify it and show more experimental data regarding cardiac involvement.

Response 6:  We have only analyzed rats which were 2.5-3 months old and at this age we have not detected any cardiomyocite cachexia or heart fibrosis. Future studies will be designed to clarify relation between p46Shc signaling and dilated cardiomyopathy. Explanation and clarification are added to manuscript at lanes 193-205.

Point 7:  The authors speculated that the connection of p52Shc/p46Shc knockout with proximal muscle weakness is likely to be explained by the sciatic nerve abnormalities. Were there any pathological changes in the cerebrum or cerebellum?

Response 7:  Sciatic nerve abnormalities in p52Shc/p46Shc knockout rats could be accompanied with pathological changes in the cerebrum or cerebellum. Detailed analysis of brain structures in our genetically modified rats is beyond the scope of this paper, which is a first report of consequences of double knockout of two Shc isoforms, which was never described in any system. It would be subject of future studies. Explanation is included at lanes 268-272.

Point 8: English language and style are fine/minor spell check required.

Response 8: Manuscript was additionally edited for language. All changes are indicated.

Round 2

Reviewer 3 Report

What did the authors use NCAM, PMCA, and WGA for?

By clarifying the reason for using these antibodies, the authors can make it easier for the reader to understand.

Author Response

Comment:  What did the authors use NCAM, PMCA, and WGA for? By clarifying the reason for using these antibodies, the authors can make it easier for the reader to understand.

Response: NCAM and PMCA are used to visualize axons since these are neuronal markers. WGA is a lectin that stains connective tissue around cells. In skeletal muscle sections we used WGA to visualize endomysium around muscle fibers. In sciatic nerve sections we used WGA to visualize endoneurium surrounding myelin sheet and centrally located axon. Clarification is added to the manuscript.